# Efficacy of PARP Inhibitor, Platinum, and Immunotherapy in BRCA-Mutated HER2-Negative Breast Cancer Patients: A Systematic Review and Network Meta-Analysis

**DOI:** 10.3390/jcm12041588

**Published:** 2023-02-17

**Authors:** Wanyi Sun, Yun Wu, Fei Ma, Jinhu Fan, Youlin Qiao

**Affiliations:** 1Department of Cancer Epidemiology, National Cancer Center, National Clinical Research Center for Cancer, Cancer Hospital, Chinese Academy of Medical Sciences and Peking Union Medical College, Beijing 100124, China; 2Department of Medical Oncology, National Cancer Center, National Clinical Research Center for Cancer, Cancer Hospital, Chinese Academy of Medical Sciences and Peking Union Medical College, Beijing 100124, China; 3Center for Global Health, School of Population Medicine and Public Health, Chinese Academy of Medical Sciences and Peking Union Medical College, Beijing 100124, China

**Keywords:** polymerase inhibitors, platinum, immunotherapy, BRCA-mutated HER2-negative breast cancer, efficacy, network meta-analysis

## Abstract

The optimal treatment regimen for breast cancer patients with gBRCA mutations remains controversial given the availability of numerous options, such as platinum-based agents, polymerase inhibitors (PARPis), and other agents. We included phase II or III RCTs and estimated the HR with 95% CI for OS, PFS, and DFS, in addition to the OR with 95% CI for ORR and pCR. We determined the treatment arm rankings by *P*-scores. Furthermore, we carried out a subgroup analysis in TNBC and HR-positive patients. We conducted this network meta-analysis using R 4.2.0 and a random-effects model. A total of 22 RCTs were eligible, involving 4253 patients. In the pairwise comparisons, PARPi + Platinum + Chemo was better than PARPi + Chemo for OS (in whole study group and in both subgroups) as well as PFS. The ranking tests demonstrated that PARPi + Platinum + Chemo ranked first in PFS, DFS, and ORR. Platinum + Chemo showed higher OS than PARPi + Chemo. The ranking tests for PFS, DFS, and pCR indicated that, except for the best treatment (PARPi + Platinum + Chemo) containing PARPi, the second and third treatments were platinum monotherapy or platinum-based chemotherapy. In conclusion, PARPi + Platinum + Chemo might be the best regime for gBRCA-mutated BC. Platinum drugs showed more favorable efficacy than PARPi in both combination and monotherapy.

## 1. Introduction

Breast cancer is the most frequent malignancy among women, with over 287,000 newly diagnosed cases and approximately 45,000 annual deaths in the United States [1]. Approximately 5% to 6% of breast cancers patients carry germline BRCA1/2 (gBRCA1/2) mutations [2,3], which are also the most common inherited mutations associated with breast cancer [4,5].

Compared with sporadic breast cancer, patients with BRCA-mutated breast cancer are characterized by a younger age of onset and higher incidences of contralateral breast cancer and second primary malignancies [3,6]. About 75% of patients with a BRCA1 mutation are disposed to triple-negative breast cancer (TNBC; estrogen-receptor (ER)-negative, progesterone-receptor (PR)-negative, and human epidermal growth factor receptor 2 (HER2)-negative), whereas the majority of BRCA2-associated breast cancers are ER-positive [7,8,9]. Although patients carrying gBRCA mutations exhibit a more aggressive tumoral behavior and increased risk for contralateral breast cancer, whether these mutations are independent prognostic factors is still uncertain [10].

BRCA1/2 are tumor-suppressor genes that encode proteins involved in DNA double-strand break repair via the homologous recombination (HR) pathway, which is a virtually error-free DNA repair mechanism [11,12]. HR is directly compromised in the event of gBRCA mutation, forcing cells to resort to alternative methods of double-strand break repair through error-prone pathways, leading to genomic instability and ultimately expediting tumorigenesis [13]. Poly (ADP-ribose) polymerase (PARP) functions in the repair of DNA single-strand breaks [14,15]. PARP inhibitors (PARPi) are synthetic lethal in cells with DNA damage defects [16] and showed promising antitumor activity in patients with BRCA-mutated cancers [17,18]. Currently, two PARP inhibitors (olaparib and talazoparib) have been approved by the Food and Drug Administration (FDA) for the treatment of gBRCA-mutated HER2-negative metastatic breast cancer.

In addition, platinum-based chemotherapeutic agents, such as cisplatin and carboplatin, are commonly used in the treatment of breast and ovarian cancers. Platinum agents, which act as DNA-damaging agents, can cause DNA strand breaks and lead to cancer cell apoptosis, which makes them more effective in cancer cells with DNA repair deficiencies caused by BRCA mutations [19]. Several trials have demonstrated that BC patients with gBRCA mutations could benefit from platinum-containing treatments [20,21,22].

HR deficiency caused by gBRCA mutations impairs the ability of DNA damage repair and confers increased sensitivity to PARPi and platinum. Previous meta-analyses have showed the comparative efficiency within platinum agents [23,24] and PARPi [25] in gBRCA breast cancer. However, determining the optimal regimen containing platinum agents, PARPis, and other agents remains controversial. Thereby, we conducted this systematic review and network meta-analysis to assess the efficacy of PARPi, platinum, and other regimens for BC patients with gBRCA mutations.

## 2. Materials and Methods

### 2.1. Design

Our systematic review and meta-analysis were exempt from institutional review board approval based on National Cancer Center criteria. We conducted our study in conformity with the preferred reporting items for systematic reviews and meta-analyses (PRISMA) recommendations. We registered our study in PROSPERO, which contains the review protocol (https://www.crd.york.ac.uk/prospero, accessed on 12 August 2022. ID: CRD42022350762). We conducted our study in strict accordance with the protocol.

### 2.2. Search Strategy

We searched databases including Pubmed, Embase, the Cochrane Library, ClinicalTrials.gov, and WHO ICTRP search portal (last search was updated on 28 March 2022). Our detailed search strategy is in Appendix A. Additionally, we further searched the bibliographies of included articles and related reviews to discover further potentially eligible research.

### 2.3. Study Selection

We considered studies meeting the following inclusion criteria eligible for inclusion in the analysis: (1) phase II or III randomized controlled trials (RCTs) including patients with breast cancer; (2) treatment with PARP inhibitor or platinum; (3) sufficient data provided about efficacy; (4) BRCA mutation result is available; and (5) if multiple publications of the same trial were retrieved, only the most recent publication was included.

Our exclusion criteria were as follows: (1) phase I clinical trial; (2) case reports, editorials, review articles, and retrospective studies; (3) single-arm, phase II study; or (4) did not report BRCA status.

### 2.4. Data Extraction

Two independent investigators reviewed the publications and extracted the data. Discrepancies were resolved by discussion. We extracted the following data: author, region, cancer stage, sample size, median age, treatment regimens, and outcomes of interest. The primary outcome of our study was overall survival (OS). Secondary outcomes included progression-free survival (PFS), disease-free survival (DFS), overall response rate (ORR), and pathologic complete response (pCR). We defined OS as the time between diagnosis and death for any cause, PFS as the time between randomization and progression or death for any reason, DFS as the time from randomization to cancer recurrence or death for any reason, ORR as the sum of partial and complete responses by the Response Evaluation Criteria in Solid Tumors, and pCR as no histological evidence of malignancy in lymph nodes in both primary and metastatic areas.

### 2.5. Data Analysis

We estimated the hazard ratio (HR) with a 95% confidence interval (CI) for OS, PFS, and DFS, and the odds ratio (OR) with 95% CI for ORR and pCR. We assessed heterogeneity by using the *I* square test (*I*^2^) and consistency by node analysis [26]. We determined the ranking of treatment arms by using *P*-Scores based on network estimates and corresponding standard errors. The *P*-score represents the rank probability of a specific rank number of different arms in every outcome (for example, PARPi’s *P*-score was the highest among all the regimens in rank number 1 in ORR, and it ranked first in ORR). The forest figure showed the results of different research outcomes compared with Chemo. We carried out a subgroup analysis in a population of TNBC and hormone receptor (HR)-positive patients. We conducted a network meta-analysis with package “meta”, R 4.2.0, (Ross Ihaka and Robert Gentleman, Auckland, New Zealand) using the random-effects model.

### 2.6. Bias Assessment

We further used Cochrane Collaboration’s tool for assessing risk of bias in randomized trials (RoB2.0, the Cochrane Methods Groups, London, UK) to evaluate the quality of the articles. We assessed publication bias via funnel plot. Two investigators independently assessed the included articles. Discrepancies were resolved by discussion.

## 3. Results

### 3.1. Search Results and Study Characteristics (Figure 1 and Figure 2, and Table 1)

We identified a total of 4253 records from the initial database search, and 2006 remained after removing duplicates. We excluded 1627 articles by reviewing titles and abstracts and by utilizing the following criteria: nonrandomized, single-arm, retrospective studies (1208), not platinum or PARPI intervention (348), and not breast cancer (71). We further excluded 356 articles by reading the full-text, and our criteria included results not yet given (343) and real-world study or retrospective study but not RCT (14). Finally, we included 22 studies in the meta-analysis. The selection progress and the main characteristics of the selected articles are displayed in Figure 1 and Table 1, respectively. Among the 22 studies, 8 were ran-domized phases 2 and 14 were randomized phase 3 trials. All included studies had two arms. We included 4332 participants. The network struc-ture diagram is shown in Figure 2. We applied network structure diagrams to display the direct association between different treatment regimens, with the thicknesses of the lines providing a measure of the number of di-rect comparisons between two regimens.

### 3.2. Overall Survival (OS) (Figure 2 and Figure 3, Table 2, and Appendix A)

The effects on OS rate were reported in ten studies, including eleven regimens. PARPi + Platinum + Chemotherapy (Chemo) was better than PARPi + Chemo (HR = 0.58, 95% CI: 0.37–0.89). Platinum + Chemo was better than PARPi + Chemo (HR = 0.67, 95% CI: 0.47–0.97). The ranking analysis based on *P*-scores suggested that, for OS, the best regimen was Atezolizumab + Chemo, followed by Platinum monotherapy, PARPi monotherapy, PARPi + Platinum + Chemo, and Chemo. The heterogeneity test showed that the *I*-square tests were less than 50%, which indicated no heterogeneity. Because of no closed loop, we could not conduct a node analysis of consistency. Figure 3 shows the HR and 95% CI of different arms in comparison with Chemo, and the results of the heterogeneity test and other details are further displayed in Appendix A, Appendix A.

### 3.3. Progression-Free Survival (PFS) (Figure 2 and Figure 3, Table 2, and Appendix A)

The effects on PFS rate were reported in eight studies, including nine regimens. PARPi + Platinum + Chemo was better than PARPi + Chemo (HR = 0.35, 95% CI: 0.11–0.99). The ranking analysis showed that the best regimen was PARPi + Platinum + Chemo, followed by Platinum monotherapy, Platinum + Chemo, PARPi monotherapy, and Atezolizumab + Chemo. The heterogeneity test showed that *I* squares were more than 50% in two comparisons (PARPi vs. Chemo and Platinum + Chemo vs. PARPi + Platinum + Chemo), which indicated heterogeneity. Our node analysis of consistency showed that the *p*-values were more than 0.05, which indicated consistency. Figure 3 shows the HR and 95% CI of different arms in comparison with Chemo, and the results of the heterogeneity test and other details are displayed in Appendix A, Appendix A.

### 3.4. Disease-Free Survival (DFS) (Figure 2 and Figure 3, Table 2, and Appendix A)

The effects on DFS rate were reported in six studies, including seven regimens. We did not find statistically significant results. Our ranking analysis suggested the best regimen was PARPi + Platinum + Chemo, followed by Platinum + Chemo, Bevacizumab + Platinum + Chemo, Chmeo, and Bevacizumab + Chemo. Our heterogeneity test showed that the *I*-square tests were less than 50%, whereas our node analysis of consistency showed that the *p*-values were more than 0.05, which indicated no heterogeneity and consistency. Figure 3 shows the HR and 95% CI of different arms in comparison with Chemo, and the results of the heterogeneity test and other details are further displayed in Appendix A, Appendix A.

### 3.5. Overall Response Rate (ORR) (Figure 2 and Figure 3, Table 2, and Appendix A)

The effects on ORR rate were reported in eight studies, including nine regimens. PARPi was better than Chemo (OR = 0.30, 95% CI: 0.12–0.80). PARPi + Platinum + Chemo was better than Chemo (OR = 0.02, 95% CI: 0.0003–0.38). Platinum + Chemo was better than Chemo (OR = 0.03, 95% CI: 0.0006–0.49). Our ranking analysis suggested that the best regimen was PARPi + Platinum + Chemo, followed by Platinum + Chemo, PARPi + Chemo, Platinum monotherapy, and PARPi monotherapy. Our heterogeneity test showed that the *I*-square tests were more than 50% in three comparisons (Platinum vs. Chemo, Platinum + Chemo vs. Chemo, and Platinum + Chemo vs. PARPi + Chemo), which indicated heterogeneity. Our node analysis of consistency showed that the *p*-values were more than 0.05, which indicated consistency. Figure 3 shows the OR and 95% CI of different arms in comparison with Chemo, and the results of the heterogeneity test and other details are further displayed in Appendix A, Appendix A.

### 3.6. Pathologic Complete Response (pCR) (Figure 2 and Figure 3, Table 2, and Appendix A)

The effects on pCR rate were reported in seven studies, including nine regimens. We did not find statistically significant results. Our ranking analysis suggested that the best regimen was Bevacizumab + Chemo, followed by Platinum monotherapy, Bevacizumab + Platinum + Chemo, PARPi + Platinum + Chemo, and Platinum + Chemo. Our heterogeneity test showed that one group (platinum vs. chemo) of *I* square was more than 50% in Platinum vs. Chemo, which indicated heterogeneity. Our node analysis of consistency showed that the *p*-values were more than 0.05, which indicated consistency. Figure 3 shows the OR and 95% CI of different arms in comparison with Chemo, and the results of the heterogeneity test and other details are further displayed in Appendix A, Appendix A.

### 3.7. Subgroup Analysis of TNBC and Hormone Receptor (HR)-Positive (Table 2 and Appendix A)

In the TNBC subgroup, for ORR, PARPi was better than Chemo (OR = 0.11, 95% CI: 0.14–0.87) and PARPi + Chemo was better than Chemo (OR < 0.01, 95% CI: 0.00–0.18). For PFS, PARPi + Platinum + Chemo was better than PARPi + Chemo (HR = 0.31, 95% CI: 0.10–0.96) and PARPi was better than Chemo (HR = 0.51, 95% CI: 0.28–0.93). In the non-TNBC subgroup, for PFS, PARPi + Platinum + Chemo was better than PARPi + Chemo (HR = 0.41, 95% CI: 0.18–0.96).

### 3.8. Publication Bias and Risk of Bias in Randomized Trials (Appendix A in Appendix A)

As for risk of bias in randomized trials, two studies indicated high risk, two studies indicated some concerns, and the other eighteen studies were low risk. For publication bias, the funnel plot of the ORR did not show asymmetry (Egger’s test *p*-value = 0.12).

## 4. Discussion

According to our results from comparing two regimens and performing a ranking analysis, we found that PARPi + Platinum + Chemo therapy might be the best regimen for HER2-negative BC patients, regardless of whether they are TNBC or HR-positive. As for the comparison of the two treatments for whole BC patients in our study, PARPi + Platinum + Chemo was better than PARPi + Chemo for OS as well as PFS, and PARPi + Platinum + Chemo was better than Chemo in ORR analysis. Furthermore, ranking tests demonstrated that PARPi + Platinum + Chemo ranked first for PFS, DFS, and ORR among all the regimens. Moreover, PARPi + Platinum + Chemo was also better than PARPi + Chemo for PFS in TNBC or HR-positive subgroups. 

The gBRCA1/2 mutation results in HRR deficiencies impart unique sensitivity to platinum-based chemotherapy and PARPi. This common mechanism might lead to cross-resistance. A study showed that the emergence of BRCA-reversion mutations was caused by a PARP inhibitor or platinum chemotherapy in BRCA1/2-mutant metastatic breast cancer [27]. However, this study testified that the combination of PARPi and platinum-based chemotherapy could benefit patients at least on efficacy. A study of phase II (BROCADE, NCT01506609) trial-enrolled patients with gBRCA-mutated advanced breast cancer showed longer median PFS and OS, with a tolerable safety profile for veliparib plus carboplatin/paclitaxel compared with carboplatin/paclitaxel alone [28,29]. More recently, the BROCADE3 phase III (NCT02163694) trial indicated that the addition of veliparib to carboplatin/paclitaxel considerably improved PFS compared with carboplatin/paclitaxel alone in patients with advanced, unresectable, HER2-negative, gBRCA-mutated breast cancer [30]. Similarly, previous studies also demonstrated the clinical benefit of adding PARPi regimens to platinum-based chemotherapy in ovarian and pancreatic cancer [31,32]. 

Additionally, this study indicated that platinum showed better efficacy than PARPi. The Platinum + Chemo regimen seemed to represent better OS than PARPi + Chemo in pairwise comparisons. The ranking tests also indicated that platinum monotherapy showed OS superior to PARPi, and Platinum + Chemo tended to show higher ORR than PARPi + Chemo. Moreover, the ranking tests for PFS, DFS, and pCR indicated that except for the best treatment (PARPi + Platinum + Chemo), which contained PARPi, the second- and the third-best treatments were platinum monotherapy or platinum-based chemotherapy.

Platinum drugs showed more favorable efficacy than PARPi in the treatment of breast cancer patients with gBRCA mutations, both in monotherapy and combining with other regimes. The possible explanation is that platinum drugs could cause DNA strand breaks by binding to it to form intra- and inter-stranded crosslinks, whereas PARPi drugs inhibited poly (ADP-ribose) polymerase from repairing DNA damage, thereby indirectly destroying DNA [16,19]. Two phase II RCTs revealed that platinum therapies displayed promising antitumor activity in metastatic TNBC, especially in patients with gBRCA1/2 mutations (TBCRC009, NCT00483223; TNT, NCT00532727) [22,33]. Few studies offered direct comparisons of platinum and PARPi. A phase II clinical trial of BROCADE-enrolled patients with gBRCA1/2-mutations-recurrent/metastatic breast cancer found that the veliparib with temozolomide group displayed slightly prolonged PFS and OS compared with placebo with carboplatin/paclitaxel, without statistical significance. However, PFS, OS, and ORR were each inferior for veliparib plus temozolomide compared with placebo plus carboplatin/paclitaxel (BROCADE, NCT01506609) [29]. Another phase II GeparOLA trial also included patients with HER2-negative breast cancer and homologous recombination deficiency and showed that paclitaxel/olaparib did not improve pCRin gBRCA TNBC patients in comparison with paclitaxel/carboplatinum (NCT02789332) [34]. Moreover, platinum’s more severe damage function, in contrast to PARPi, could also be revealed through adverse reaction. The BROCADE study indicated that the PARPi regimen provided tolerability advantages over platinum, with less frequent neutropenia, anemia, alopecia, and neuropathy [29]. Therefore, the platinum regimen exhibits better efficacy but the PARPi regimen might be more suitable for patients with poor general conditions.

In our study, the Atezolizumab + Chemo regimen ranked first regarding OS. Tumor immune microenvironment also played a considerable role in the occurrence and development of tumors in BC patients, especially in the PD-L1-positive subgroup [35,36,37]. Furthermore, immunotherapy exhibited a considerable tailing effect, indicating that it might work better than conventional chemotherapy in some measures of long-term effects, such as OS [38,39]. In our research, there was only one study that compared Atezolizumab + Chemo versus Chemo. The IMpassion 130 phase III trials showed a numerical but insignificant improvement in PFS (median: 7.4 vs. 5.5 months; HR = 0.69, 95% CI: 0.42–1.12) and OS (median: 28.9 vs. 20.1 months; HR = 0.71, 95% CI: 0.39–1.29) in gBRCA-mutated TNBC patients treated with atezolizumab and nab-paclitaxel compared with placebo and nab-paclitaxel [40]. Further studies are needed to explore the efficacy of immunotherapy combination treatments among gBRCA-mutated BC patients.

Bevacizumab combined with chemotherapy ranked higher than other regimens in pCR. Previous studies have verified that patients with gBRCA mutations tend to overexpress vascular endothelial growth factor (VEGF), angiopoietin (Ang)-1, and Ang-2 [41]. These findings supported the application of bevacizumab to inhibit angiogenesis and induce hypoxia-related DNA damage, which may contribute to synthetic lethal reactions in tumors with BRCA1/2 mutations. Our analysis found that bevacizumab considerably improved the pCR rate; however, in the P-score ranking results of DFS, Bevacizumab + Chemo was even inferior to Chemo alone, although the difference was not statistically significant. Based on the survival outcomes, Bevacizumab + Chemo should not be recommended in gBRCA breast cancer patients.

There are also some important ongoing RCTs in patients with gBRCA1/2-mutated BC. In one study, the novel PARPi fluzoparib plus apatinib, a VEGFR2 tyrosine kinase inhibitor, is oriented for patients with gBRCA-mutated HER2-negative metastatic BC (NCT04296370). Moreover, olaparib combined with the PD-1/PD-L1 inhibitors, such as pembrolizumab (NCT04191135), atezolizumab (NCT02849496), and durvalumab (NCT03167619), is currently being tested for the treatment of gBRCA1/2-mutated BC. In addition, niraparib combined with aromatase inhibitors is in trials for gBRCA luminal-like BC (NCT04240106). Following the publication of clinical trial results, these findings may lead to striking changes in the treatment patterns for gBRCA-mutated BC, helping physicians to select more prudent therapeutic options.

In general, this systematic review and network meta-analysis is based on high-quality RCTs, which may offer solid evidence to guide clinical practice for breast cancer patients with gBRCA mutations. Inevitably, several limitations existed in this study. First, the limited network connectivity due to the small number of studies leads to large confidence intervals for some estimates, even when effect sizes are large. Second, various cytotoxic agents, such as anthracyclines and alkylating agents, still have DNA-damaging effects. The diversity of cytotoxic agents increased the heterogeneity of the Chemo arm, which makes it difficult to eliminate in this study. Additionally, clinical dosages of different trials also contributed to variations in the result, which needs further analysis if there are enough eligible studies in the future. Thirdly, the *p*-values may only reflect the intervention rankings, lacking the ability to present the degree of absolute difference. Given patient heterogeneity, the superior efficacy observed should be viewed with some caution, particularly when considering applying these findings to clinical practice.

## 5. Conclusions

In conclusion, we found that, among HER2-negative BC patients, regardless of whether they were TNBC or HR-positive patients with gBRCA mutations, PARPi + Platinum + Chemo therapy might be the best regimen. As for the efficacy, this study indicated that platinum might be better than PARPi. Following the OS results, future studies should contain more variations in immunotherapy to explore its efficacy in gBRCA-mutated BC patients. Our main results indicate that platinum-based chemotherapy associated with PARP inhibitors might be recommended for HER2-negative BC patients with gBRCA mutations. Additionally, when considering monotherapy combined with Chemo, platinum is better than a PARP inhibitor. Further head-to-head randomized clinical trials are needed to directly compare the efficacy of platinum and PARP inhibitors in BRCA-mutated breast cancer. 

## Figures and Tables

**Figure 1 jcm-12-01588-f001:**
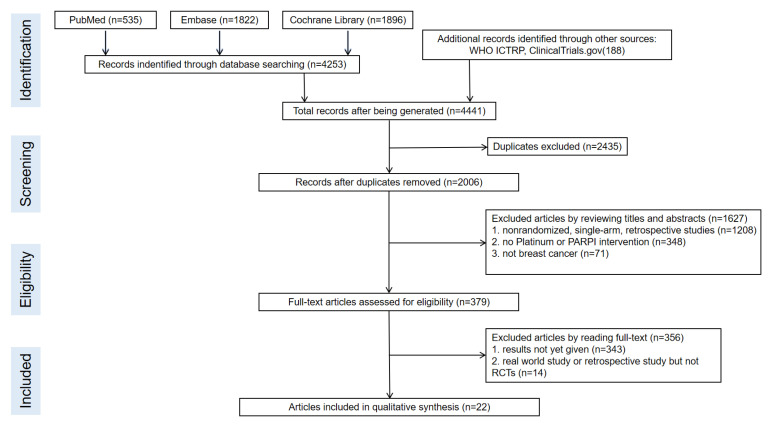
Flow chart of the article selection process.

**Figure 2 jcm-12-01588-f002:**
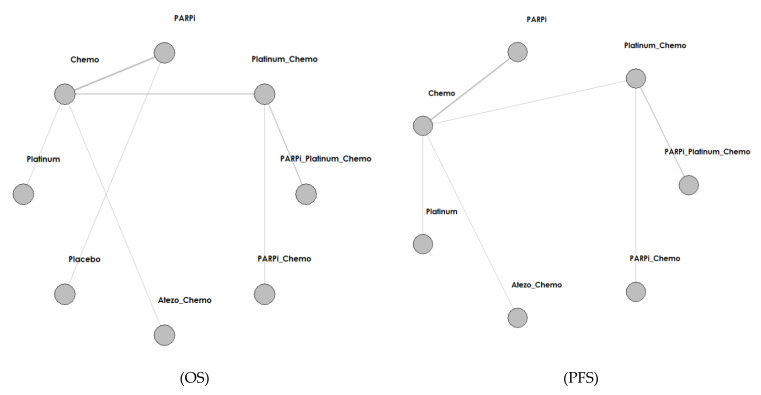
Network structure diagram according to different outcomes. Abbreviations: PARPi for PARP inhibitor, Bev for bevacizumab, Atezo for atezolizumab, Chemo for chemotherapy.

**Figure 3 jcm-12-01588-f003:**
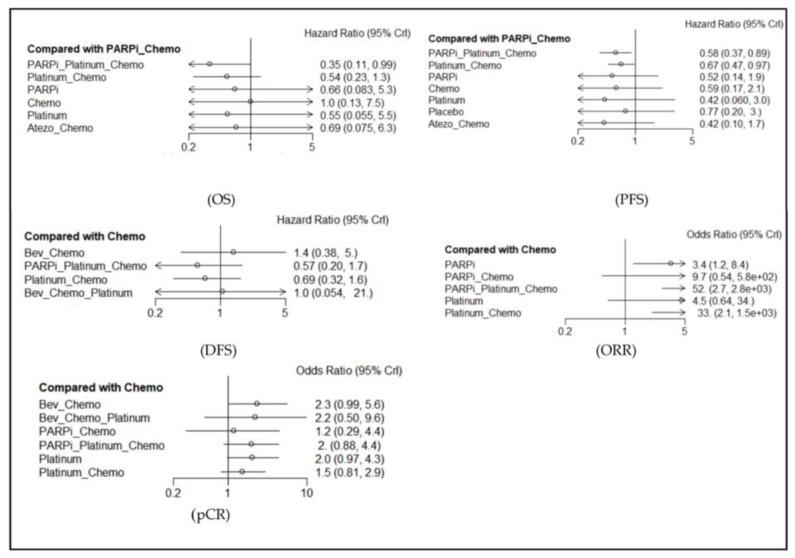
Forest map of main results (in comparison with chemotherapy). Abbreviations: PARPi for PARP inhibitor, Bev for bevacizumab, Atezo for atezolizumab, Chemo for chemotherapy.

**Table 1 jcm-12-01588-t001:** Characteristics of articles included in meta-analysis.

Study	Study Abbr.	Phase	Region	Stage	Patients	Treatment Regimen	Sample Size	Median Age (Range)
Byrski, T. 2010	NA	2	Canada	Neoadjuvant	HER2-negative BC	Platinum (Cisplatin)	12	43 (37–50)
						Chemo (Doxorubicin + Cyclophosphamide)	90	42 (25–50)
Arun, B. K. 2021	BROCADE3	3	Multicenter	Advanced	HER2-negative BC	PARPi + Platinum + Chemo (Veliparib + Carboplatin + Paclitaxel)	274	47 (24–82)
						Platinum + Chemo (Placebo + Carboplatin + Paclitaxel)	139	44 (28–75)
Han, H. S. 2018	BROCADE	2	Multicenter	Advanced	HER2-negative BC	PARPi + Platinum + Chemo (Veliparib + Carboplatin + Paclitaxel)	95	44 (25–65)
						PARPi + Chemo (Veliparib + Temozolomide)	91	46 (22–70)
						Platinum + Chemo (Placebo + Carboplatin + Paclitaxel)	98	44 (24–66)
Du, F. 2020	NA	2	China	Adjuvant	TNBC	Platinum + Chemo (Docetaxel or Paclitaxel + Carboplatin)	6	NA
						Chemo (Epirubicin + Cyclophosphamide, followed by Docetaxel or Paclitaxel)	9	NA
Fasching, P. A. 2021	GeparOLA	2	German	Neoadjuvant	HER2-negative BC	PARPi + Chemo (Olaparib + Paclitaxel)	41	48 (25–71)
						Platinum + Chemo (Carboplatinum + Paclitaxel)	18	45 (26–67)
Fasching, P. A. 2018	GeparQuinto	3	German	Neoadjuvant	TNBC	Bev + chemo (Bevacizumab + Epirubicin + Cyclophosphamide, followed by Docetaxel)	39	48 (37–59)
						Chemo (Epirubicin + Cyclophosphamide, followed by Docetaxel)	51	48 (38–58)
Geyer, C. E. 2022	BrightTNess	3	Multicenter	Neoadjuvant	TNBC	PARPi + Platinum + Chemo (Veliparib + Carboplatin + Paclitaxel)	46	51 (41–59)
						Platinum + Chemo (Placebo + Carboplatin + Paclitaxel)	24	49 (40–57)
						Chemo (Paclitaxel)	22	50 (42–59)
Hahnen, E. 2017	GeparSixto	3	German	Neoadjuvant	TNBC	Bev + chemo + Platinum (Bevacizumab + Paclitaxel + Doxorubicin + Carboplatin)	26	48 (37–59)
						Bev + chemo (Paclitaxel + Doxorubicin + Carboplatin)	24	48 (37–59)
Robson, M. 2019	OlympiAD	3	Multicenter	Advanced	HER2-negative BC	PARPi (Olaparib)	205	41.5
						Chemo (Capecitabine, Vinorelbine, or Eribulin)	97	42
Kalra, M. 2021	NA	3	USA	Adjuvant	TNBC	PARPi + Platinum (Rucaparib + Cisplatin)	8	47 (21–75)
						Platinum (Cisplatin)	14	48 (27–69)
Kummar, S. 2016	NA	2	USA	Advanced	TNBC	PARPi + Chemo (Veliparib + Cyclophosphamide)	2	54 (34–77)
						Chemo (Cyclophosphamide)	5	54 (34–77)
Litton, J. K. 2020	EMBRACA	3	USA	Advanced	HER2-negative ABC	PARPi (Talazoparib)	219	45 (27–84)
						Chemo (Capecitabine, Eribulin, Gemcitabine, or Vinorelbine)	114	50 (24–88)
Zhang, J. 2018	CBCSG006	3	China	Advanced	TNBC	Platinum + Chemo (Cisplatin + Gemcitabine)	6	NA
						Chemo (Paclitaxel + Gemcitabine)	8	NA
Pohl-Rescigno, E. 2020	GeparOcto	3	Multicenter	Neoadjuvant	HER2-negative BC	Platinum + Chemo (Carboplatin + Paclitaxel + Doxorubicin)	47	48 (21–76)
						Chemo (Epirubicin + Aclitaxel + Cyclophosphamide)	49	48 (21–76)
Tung, N. 2020	INFORM	2	USA	Neoadjuvant	HER2–negative BC	Platinum (Cisplatin)	60	42 (24–73)
						Chemo (Doxorubicin-Cyclophosphamide)	58	42 (24–73)
Turner, N. C. 2021	BRAVO	3	Multicenter	Advanced	HER2–negative BC	PARPi (Niraparib)	141	NA
						Chemo (Eribulin, Capecitabine, Vinorelbine, or Gemcitabine)	74	NA
Tutt, A. 2018	TNT	3	UK	Advanced	TNBC	Platinum (Carboplatin)	25	56 (48–63)
						Chemo (Docetaxel)	18	55 (48–63)
Tutt, A. N. J. 2021	OlympiA	3	Multicenter	Adjuvant	HER2–negative BC	PARPi (Olaparib)	921	43 (33–53)
						Placebo (Placebo)	915	44 (33–53)
Yu, K. D. 2020	PATTERN	3	China	Adjuvant	TNBC	Platinum + Chemo (Carboplatin + Paclitaxel)	34	51 (44–57)
						Chemo (Cyclophosphamide, Epirubicin, and Fluorouracil, followed by Docetaxcel)	32	51 (44–57)
Zheng, F. 2022	NA	2	China	Adjuvant	TNBC	Platinum + Chemo (Carboplatin + Paclitaxel + Docetaxel)	12	48 (43–54)
						Chemo (Epirubicin, Cyclophosphamide, followed by Docetaxel or Paclitaxel)	26	47 (42–56)
Emens, L. A. 2021	IMpassion130	3	Multicenter	Advanced	TNBC	Atezo + Chemo (Atezolizumab + Nab-paclitaxel)	39	NA
						Chemo (Nab-paclitaxel)	50	NA
Sella, T. 2018	NA	2	Israel	Neoadjuvant	TNBC	Platinum + Chemo (Anthracycline + Cyclophosphamide, followed by Paclitaxel + Carboplatin)	14	42
						Chemo (Anthracycline + Cyclophosphamide)	34	43

Abbreviations: NA for not available, HER2 for human epidermal growth factor receptor 2, PARPi for PARP inhibitor, Bev for bevacizumab, Atezo for atezolizumab, Chemo for chemotherapy.

**Table 2 jcm-12-01588-t002:** Meta-analysis of efficacy for different therapies.

OS (Hazard Ratios (HR) with 95% CI)							
PARPi + Platinum + Chemo							
0.85 (0.67, 1.09)	Platinum + Chemo						
1.11 (0.32, 3.87)	1.30 (0.38, 4.44)	PARPi					
0.97 (0.28, 3.37)	1.14 (0.34, 3.85)	0.88 (0.74, 1.04)	Chemo				
1.37 (0.19, 9.62)	1.6 (0.23, 11.08)	1.24 (0.27, 5.58)	1.40 (0.31, 6.27)	Platinum			
0.75 (0.20, 2.83)	0.88 (0.24, 3.24)	0.68 (0.44, 1.05)	0.77 (0.48, 1.24)	0.55 (0.12, 2.65)	Placebo		
1.37 (0.35, 5.45)	1.61 (0.42, 6.24)	1.24 (0.67, 2.31)	1.41 (0.77, 2.56)	1.00 (0.20, 5.06)	1.82 (0.85, 3.90)	Atezo + Chemo	
**0.58 (0.37, 0.89)**	**0.67 (0.47, 0.97)**	0.52 (0.14, 1.87)	0.59 (0.17, 2.10)	0.42 (0.06, 3.02)	0.76 (0.20, 2.95)	0.42 (0.10, 1.70)	PARPi + Chemo
PFS (hazard ratios (HR) with 95% CI)							
PARPi + Platinum + Chemo							
0.64 (0.33, 1.19)	Platinum + Chemo						
0.54 (0.07, 3.89)	0.84 (0.12, 5.54)	PARPi					
0.35 (0.05, 2.44)	0.55 (0.09, 3.44)	0.65 (0.41, 1.10)	Chemo				
0.64 (0.07, 5.98)	1.00 (0.12, 8.64)	1.20 (0.36, 4.11)	1.83 (0.60, 5.59)	Platinum			
0.51 (0.06, 4.29)	0.80 (0.10, 6.14)	0.95 (0.34, 2.77)	1.46 (0.58, 3.66)	0.80 (0.19, 3.38)	Atezo + Chemo		
**0.35 (0.11, 0.99)**	0.54 (0.23, 1.29)	0.64 (0.08, 5.17)	0.98 (0.13, 7.37)	0.54 (0.05, 5.39)	0.67 (0.07, 6.19)	PARPi + Chemo	
DFS (hazard ratios (HR) with 95% CI)							
Bev + Chemo							
1.38 (0.38, 5.03)	Chemo						
2.42 (0.44, 12.62)	1.75 (0.58, 5.00)	PARPi + Platinum + Chemo					
2.00 (0.42, 8.80)	1.44 (0.61, 3.16)	0.82 (0.27, 2.39)	Platinum + Chemo				
1.34 (0.09, 19.90)	0.96 (0.05, 18.96)	0.55 (0.02, 13.15)	0.67 (0.03, 14.72)	Bev + Chemo + Platinum			
pCR (odds ratios (OR) with 95% CI)							
Bev + Chemo							
1.06 (0.32, 3.56)	Bev + Chemo + Platinum						
2.32 (0.99, 5.59)	2.19 (0.50, 9.56)	Chemo					
2.02 (0.41, 10.28)	1.90 (0.26, 14.24)	0.86 (0.23, 3.45)	PARPi + Chemo				
1.19 (0.37, 3.90)	1.12 (0.21, 6.00)	0.51 (0.23, 1.14)	0.59 (0.14, 2.43)	PARPi + Platinum + Chemo			
1.14 (0.37, 3.61)	1.08 (0.20, 5.61)	0.49 (0.23, 1.04)	0.57 (0.12, 2.62)	0.96 (0.32, 2.88)	Platinum		
1.53 (0.52, 4.53)	1.44 (0.29, 7.23)	0.66 (0.34, 1.25)	0.76 (0.22, 2.44)	1.28 (0.58, 2.83)	1.34 (0.50, 3.56)	Platinum + Chemo	
ORR (odds ratios (OR) with 95% CI)							
Chemo							
**0.30 (0.12, 0.80)**	PARPi						
0.10 (0.001, 1.91)	0.34 (0.004, 7.12)	PARPi + Chemo					
**0.02 (0.0003, 0.38)**	0.06 (0.0009, 1.40)	0.18 (0.03, 1.76)	PARPi + Platinum + Chemo				
0.22 (0.03, 1.57)	0.74 (0.08, 6.34)	2.20 (0.06, 233)	11.93 (0.32, 1163)	Platinum			
**0.03 (0.0006, 0.49)**	0.10 (0.002, 1.87)	0.28 (0.07, 1.97)	1.58 (0.50, 5.41)	0.13 (0.002, 4.26)	Platinum + Chemo		
PFS of TNBC subgroup (hazard ratios (HR) with 95% CI)							
PARPi + Platinum + Chemo							
0.76 (0.41, 1.43)	Platinum + Chemo						
0.81 (0.11, 5.93)	1.07 (0.16, 7.17)	PARPi					
0.41 (0.06, 2.78)	0.54 (0.09, 3.36)	**0.51 (0.28, 0.93)**	Chemo				
0.74 (0.08, 6.88)	0.98 (0.12, 8.35)	0.92 (0.26, 3.22)	1.81 (0.61, 5.48)	Platinum			
0.60 (0.07, 4.89)	0.79 (0.10, 5.94)	0.73 (0.25, 2.19)	1.46 (0.59, 3.59)	0.80 (0.19, 3.29)	Atezo + Chemo		
**0.31 (0.10, 0.96)**	0.41 (0.16, 1.04)	0.39 (0.05, 3.15)	0.76 (0.10, 5.79)	0.42 (0.04, 4.13)	0.52 (0.06, 4.73)	PARPi + Chemo	
PFS of hormone receptor (HR)-positive subgroup (hazard ratios (HR) with 95% CI)							
PARPi + Platinum + Chemo							
0.67 (0.42, 1.07)	Platinum + Chemo						
**0.41 (0.18, 0.96)**	0.61 (0.30, 1.24)	PARPi + Chemo					
ORR of TNBC subgroup (odds ratios (OR) with 95% CI)							
Chemo							
**0.11 (0.01, 0.87)**	PARPi						
**0.000 (0.000, 0.18)**	0.000 (0.000, 1.95)	PARPi + Chemo					
0.22 (0.01, 3.87)	2.06 (0.06, 65.41)	3.86 (0.89, 75.18)	Platinum				
0.08 (0.001, 2.74)	0.71 (0.01, 40.78)	2.83 (0.18, 58.20)	0.34 (0.002, 33.32)	Platinum + Chemo			

## Data Availability

The datasets used and/or analyzed during the current study are available from the corresponding author upon reasonable request.

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
