# Peer review of "Efficacy of PARP Inhibitor, Platinum, and Immunotherapy in BRCA-Mutated HER2-Negative Breast Cancer Patients: A Systematic Review and Network Meta-Analysis"

_jcm, 2023, doi:10.3390/jcm12041588_

Round 1

Reviewer 1 Report

The authors evaluated the effectiveness of each treatment and its association with effective and optimal treatment of breast cancer patients with gBRCA mutations, based on cases in reported papers. Despite the limited sampling and many presumptive conditions, the methods used in this study are scientifically appropriate and may be valid for the findings obtained. However, although the abstract and conclusions state that platinum preparations directly damage DNA, I understand that this has not been confirmed by the authors and is still a matter of conjecture and debate. Therefore, at this stage, it should be used as appropriate in this paper.

Author Response

Response: Thank you for your positive evaluation.

We totally agree that platinum agents, acted as DNA damaging agents, can cause DNA strand breaks and lead to tumor cell apotosis. However, it still remains controversial whether platinum directly damages DNA. According to your advice, we have revised our abstract and discussion(lines 33-34 on page 1 and lines 384-386 on page 15).

Reviewer 2 Report

Recently similar work has been reported by some researchers, I am sharing links below:

https://www.ncbi.nlm.nih.gov/pmc/articles/PMC8215282/

https://www.frontiersin.org/articles/10.3389/fonc.2021.742139/full

Since the publication time to above manuscripts is too small for a study to report something novel or provide updated information on the theme. Additionally, manuscript needs extensive editing and seems to have been crudely done. Captions need to be elaborated and risk scores etc should be incorporated in the Forest plots as well as other information as mentioned in manuscripts shared above.

Whole manuscript needs a redo and although authors try to combine three treatments and do network analysis, the theme has already been extensively studied.

Author Response

Response: Thanks for the insightful comment.

Recently, the growing success of the new drugs, especially promising PARP inhibitors, has changed the treatment landscape for patients with gBRCA mutations BC. However, the optimal regimen of gBRCA mutations BC patients needs to be explored. PARP inhibitors, which have shown promising efficacy in some randomized controlled phase III trials, are still lacking direct comparison with platinum agents and other combination therapy. Therefore, we conducted this network meta-analysis especially based on gBRCA mutations BC group, in the hope of providing a reliable reference for the treatment of those patients and the design of clinical trials in the future.

Moreover, after careful reading of references you presented as similar work, we contend that our study still represented different interesting fields and held significant results.

https://www.ncbi.nlm.nih.gov/pmc/articles/PMC8215282/This study was a traditional meta-analysis only based on two arms comparison (PARP inhibitor versus chemotherapy) and mainly on BC patients. However, our study, using network meta-analysis, tried to summarize the best regimen among all possible clinical drugs used specially on gBRCA mutations BC patients.

https://www.frontiersin.org/articles/10.3389/fonc.2021.742139/full The study only focused on advanced TNBC, without making analysis on early stages such as the adjuvant or neoadjuvant patients, whereas our study included different stages of gBRCA mutations BC patients. Besides, it was also a traditional meta-analysis comparing the efficacy of PARP inhibitor with control group.

https://link.springer.com/content/pdf/10.1186/s40001-022-00839-0.pdf This study was a meta-analysis comparing the efficacy of platinum-containing and non-platinum-containing regimes on TNBC patients. It did not focus on the gBRCA population, nor did it include PRAPi, which has recently made significant progress in the treatment of TNBC.

In summary, our bayesian network meta-analysis was conducted to identify the optimal treatment and compare the efficacy of PARPi, platinum and other combination therapies for BC patients with gBRCA mutations.

Response: Thanks for your careful work.

We have revised our captions of table and figure in our manuscripts and appendix for clarity (Figure2, Figure3, Table1 and Table2 in manuscript, network structure diagrams of subgroup in Appendix A). Besides, we further provided forest plots incorporating all the network arms to show details in Figure A3-A11, Appendix A (Page 17-22).

Additionally, we have revised the entire manuscript very carefully and all the grammar errors have been corrected in the revised manuscript. Furthermore, we have had the manuscript polished in writing with MDPI English language editing service (English-edited-59079).

Reviewer 3 Report

I only have a few points to add:

1) Please clearly indicate the exclusion criteria below the inclusion criteria in the methodology.

2)Although there is no major issue with the language or information flow, there are several grammatical errors, typos, and incomplete sentences in the manuscript. I recommend that you have a native English speaker, or a professional language editing service go through the manuscript carefully. 

3)Finally, please change "The optimal regimen containing platinum, polymerase inhibitors (PARPis) and other 16 agents still remain controversial among breast cancer (BC) patients with gBRCA mutations" to "The optimal treatment regimen for breast cancer patients with gBRCA mutations remain controversial, given the availability of numerous options, such as platinum-based agents, polymerase inhibitors (PARPis) and  16 other agents"

Author Response

Response: Thank you for your advice.

We have added inclusion criteria in materials and methods according to the reviewer’s suggestion(lines 108-110 on page 3).

We have revised the entire manuscript very carefully and all the grammar errors have been corrected in the revised manuscript. Furthermore, we have had the manuscript polished in writing with MDPI English language editing service (English-edited-59079).

We have revised our abstract according to the reviewer’s suggestion(lines 16-20 on page 1).

Reviewer 4 Report

Summary: Breast cancer patients with germline mutation of BRCA1/2 (gBRCA1/2) tend to be more aggressive. Previous studies have focused on evaluate the therapeutic effect of platinum agents and PARP inhibitors (PARPi) in patients with gBRCA1/2. However, optimized therapeutic regime has not been determined. In this manuscript, the authors are aiming to systematically determine the best therapeutic regime for breast cancer patients with gBRCA1/2. Overall, they found that PARPi+ Platinum+Chem therapy might be the best regimen for HER2-negative BC patients with gBRCA mutations regardless of TNBC or HR-positive patients.

Although the small number of studies, the systemic analysis could provide a significant treatment strategy for breast cancer patients, especially for HER2-negative with gBRCA1/2 mutated BC patients.

1.     As mentioned in the discussion, there are different level of cytotoxicity in different DNA damage agents. Results of various combination of DNA damage agents with Chemotherapy may still vary. Clinical dosage may also contribute to the variation of the result.

2.     It was not too clear that immunotherapy combination (Atezolizumab+Chemo or others), was a good therapeutic regimen or else?

3.     Not sure how the P-score generated?

Author Response

Response: Thank you for your precise comments.

We totally agreed that different DNA damage agents such as alkylating agents, platinum and anthracyclines cause various levels of cytotoxicity, and also the combination of them with chemotherapy will also vary in efficacy. We have discussed this proportion of content and referred as one of the limitations in the Discussion section (lines 388-394 on page 16). Regretfully, for drawing rigorous conclusion, we admitted our limitation in discussion since we didn’t make further analysis according clinical dosage, owing to the restricted number of inclusion trials (lines 458-460 on page 18).

In our study, atezolizumab+chemo regimen ranked first regarding OS for patients with gBRCA mutation. However, there was only one study compared Atezolizumab+Chemo versus Chemo (IMpassion130: NCT02425891), which showed a numerical but insignificant improvement in PFS (median: 7.4 vs. 5.5 months; HR=0.69, 95%CI: 0.42-1.12) and OS (median: 28.9 vs. 20.1 months; HR=0.71, 95%CI: 0.39-1.29) in gBRCA-mutated TNBC patients treated with atezolizumab and nab-paclitaxel compared to placebo and nab-paclitaxel. Thereby, further studies are needed to explore the efficacy of immunotherapy combination treatments among gBRCA mutations BC patients. We have revised our discussion to offer further explanation according to the reviewer’s suggestion(lines 421-425 on page 17).

We have revised materials and methods to offer further explanation according to the reviewer’s suggestion(lines 129-132 on page 3).